# Revisiting the Injury Mechanism of Goat Sperm Caused by the Cryopreservation Process from a Perspective of Sperm Metabolite Profiles

**DOI:** 10.3390/ijms25169112

**Published:** 2024-08-22

**Authors:** Chunyan Li, Chunrong Lv, Allai Larbi, Jiachong Liang, Qige Yang, Guoquan Wu, Guobo Quan

**Affiliations:** 1Yunnan Animal Science and Veterinary Institute, Jindian, Panlong District, Kunming 650224, China; chunyanli2023@163.com (C.L.); chunronglv228@163.com (C.L.); ljj200311@163.com (J.L.); wuguoquan1982@163.com (G.W.); 2Yunnan Provincial Engineering Research Center of Livestock Genetic Resource Conservation and Germplasm Enhancement, Jindian, Panlong District, Kunming 650224, China; 3National Regional Genebank (Yunnan) of Livestock and Poultry Genetic Resources, Jindian, Panlong District, Kunming 650224, China; 4Laboratory of Sustainable Agriculture Management, Higher School of Technology Sidi Bennour, Chouaib Doukkali University El Jadida, El Jadida 24000, Morocco; allay.larbi@gmail.com; 5College of Veterinary Medicine, Yunnan Agricultural University, Fengyuan Road, Panlong District, Kunming 650500, China; yqgppy@163.com

**Keywords:** sperm, cryopreservation, cryoinjury, metabolome, bioinformatics, metabolite

## Abstract

Semen cryopreservation results in the differential remodeling of the molecules presented in sperm, and these alterations related to reductions in sperm quality and its physiological function have not been fully understood. Given this, this study aimed to investigate the cryoinjury mechanism of goat sperm by analyzing changes of the metabolic characteristics in sperm during the cryopreservation process. The ultra-high-performance liquid chromatography–quadrupole time-of-flight mass spectrometry (UHPLC-QTOF-MS) technique was performed to explore metabolite profiles of fresh sperm (C group), equilibrated sperm (E group), and frozen–thawed sperm (F group). In total, 2570 metabolites in positive mode and 2306 metabolites in negative mode were identified, respectively. After comparative analyses among these three groups, 374 differentially abundant metabolites (DAMs) in C vs. E, 291 DAMs in C vs. F, and 189 DAMs in E vs. F were obtained in the positive mode; concurrently, 530 DAMs in C vs. E, 405 DAMs in C vs. F, and 193 DAMs in E vs. F were obtained in the negative mode, respectively. The DAMs were significantly enriched in various metabolic pathways, including 31 pathways in C vs. E, 25 pathways in C vs. F, and 28 pathways in E vs. F, respectively. Among them, 65 DAMs and 25 significantly enriched pathways across the three comparisons were discovered, which may be tightly associated with sperm characteristics and function. Particularly, the functional terms such as TCA cycle, biosynthesis of unsaturated fatty acids, sphingolipid metabolism, glycine, serine and threonine metabolism, alpha-linolenic acid metabolism, and pyruvate metabolism, as well as associated pivotal metabolites like ceramide, betaine, choline, fumaric acid, L-malic acid and L-lactic acid, were focused on. In conclusion, our research characterizes the composition of metabolites in goat sperm and their alterations induced by the cryopreservation process, offering a critical foundation for further exploring the molecular mechanisms of metabolism influencing the quality and freezing tolerance of goat sperm. Additionally, the impacts of equilibration at low temperature on sperm quality may need more attentions as compared to the freezing and thawing process.

## 1. Introduction

Semen cryopreservation is a long-term preservation protocol of male germplasm resources that enables the improvement of genetic gain via artificial insemination in the absence of paternal individuals [1,2]. So far, with the advancement of semen cryopreservation technology, this practice has been ongoing over 70 years [3,4]. Nonetheless, the freezing and thawing process has been shown to cause considerable biophysical and biochemical changes in sperm, resulting in reduced sperm quality, which is mainly manifested in the impairment of its motility, plasma membrane and acrosome integralities, antioxidative enzyme activities, as well as mitochondrial function, etc. [5,6,7]. These alterations adversely affect post-thaw sperm’s capacity to fertilize and subsequent embryo development [8,9].

Mature sperm belongs to the terminal cell that lacks transcriptional and translational activities. Thus, once sperm leaves the male reproductive tract, post-translational modification and metabolic regulation play crucial roles in determining sperm viability and specific function before fertilizing an oocyte [10,11]. Up to now, there has been a series of reports about sperm metabolic regulation in vitro; for instance, glycolysis and oxidative phosphorylation are two key metabolic pathways that provide energy in the form of ATP [12,13]. TCA cycle, a central route of oxidative phosphorylation, is necessary for biosynthesis and redox homeostasis [14,15], while cAMP-PKA signaling is involved in mitochondrial dynamics and sperm motility [16,17]. At the same time, it is also important that lipid remodeling of the plasma membrane occurs during sperm capacitation [18]. In recent years, advanced omics technologies have been incorporated into studying sperm physiology and metabolic regulation [19]. For instance, metabolomics is a relatively young branch of omics science that aims to delineate the molecular profiling of products at different stages from metabolic reaction and cellular processes, which mainly focus on hydrophilic small molecules [20]. At present, some semen-derived compounds such as L-carnitine, butanoic acid and phosphatidylcholine (33:2/36:2), have been identified by the above omics technologies, and could be utilized as biomarkers for male fertility [19,21].

In this study, we integrated the metabonomics approach to explore metabolic characteristics of sperm during the freezing process. The investigation will encompass fresh sperm, equilibrium sperm and frozen sperm samples, collected from *Yunshang black* goat, a newly recognized superior breed in China. Our goal is to investigate the association between sperm and compounds’ composition, and we expect to identify potential biological markers of frozen–thawed sperm quality, thereby contributing to deepening the understanding of the metabolic mechanisms of sperm during cryopreservation. 

## 2. Results

### 2.1. Metabolic Findings of Sperm Samples Using UHPLC-QTOF-MS

In our study, a comprehensive metabolomic analysis was performed on sperm samples from the three experimental groups using an untargeted metabolomics approach with UHPLC-QTOF-MS. After QC and filter peaks computing, we identified 2570 metabolites in the positive (POS) and 2306 metabolites in the negative (NEG) mode (Appendix A). PCA was employed to assess QC; each group was represented with a different color. PC1 explained <27% of variation, PC2 explained <15% of variation, and PC3 explained <11% of variation. Three groups in the PCA diagram are delineated, and the indicated data were normal (Figure 1).

### 2.2. Identification and Functional Annotation of Differential Metabolites

Differentially abundant metabolites (DAMs) between groups were identified using variable importance in projection (VIP) score >1 in OPLS-DA models and a Student’s *t*-test (*p* value < 0.05). After comparative analyses, 374 DAMs in C vs. E, 291 DAMs in C vs. F, and 189 DAMs in E vs. F were obtained in the positive mode; concurrently, 530 DAMs in C vs. E, 405 DAMs in C vs. F, and 193 DAMs in E vs. F were obtained in the negative mode, respectively (Appendix A, Figure 2). An HMDB database (https://hmdb.ca/ (accessed on 18 August 2024)) [22] search and KEGG enrichment analysis showed these DAMs were significantly enriched in various metabolic pathways, including 31 pathways in C vs. E, 25 pathways in C vs. F, and 28 pathways in E vs. F, respectively (Appendix A), with specific pathways and the enrichment degree differing among the three comparisons (Figure 3). Among them, 65 DAMs and 25 significantly enriched biological pathways across the three comparisons were discovered that may be tightly related to sperm characteristics and function (Appendix A). Additionally, the above information also implies that the impacts of equilibration at low temperature on sperm quality may need more attention as compared to the freezing and thawing process. 

### 2.3. Classification and Hierarchical Clustering of Significantly DAMs

Distribution profiles of 65 significant DAMs among three comparative groups were constructed using the Mfuzz clustering software (v2.52.0). These significant DAMs were classified into 12 clusters based on their abundance patterns across three sample groups, as illustrated in Figure 4. The corresponding details of these DAMs are shown in Appendix A. In addition, it is well known that clustering analysis can aggregate similar substances together and divide different substances into different categories. The clustering heat map obtained via clustering analysis can directly reflect the clustering and distribution of data, and the height or density levels of data can be inferred from the gradient of colors in the clustering heat map. Thus, we performed hierarchical clustering for the above significant DAMs using the Complex-Heatmap software (v2.12.0). All significant DAMs were filtered for all categories which were enriched in at least one of the clusters with *p* < 0.05. The filtered *p* value matrix was transformed by the function x = −log10, x values were Z-transformed, and Z scores were then clustered by one-way hierarchical clustering. Another visualized heat map of the above significant DAMs is shown in Figure 5. These DAMs were mainly categorized into 2-search-DB, 2-MetDNA, and 2-DCA classes. 

## 3. Discussion

Semen cryopreservation is one of the more effective strategies for the long-term genetic resource preservation of individuals, which makes a substantial contribution to the advancement in reproductive biology. Such a strategy is important for the maintenance of endangered animals, human medical research, and livestock breeding via artificial insemination in animal husbandry [23,24]. However, freeze–thawed sperm may lose some of its fertility as a result of osmotic-induced necrosis during the freezing and thawing process, as well as oxidative injuries due to elevated ROS levels generation [5,25]. The quality of the sperm, such as its motility, membrane integrity, acrosome integrity, and DNA integrity [6,26], may be influenced by a variety of variables such as extenders composition, cryopreservation steps, and species [23]. To date, the specific mechanisms of these influences are not fully understood. 

It is well known that sperm is a highly specialized cell, and its transcriptional and translational activities become inactive once it leaves the male reproductive tract. It mainly relies on the communication and interaction of a fixed population of proteins and metabolites in order to ensure its specific functions before fertilizing an oocyte [10,11]. We have identified changes in the protein profiles of ram sperm during the freezing–thawing process [27]. This study concentrated on the metabolome alteration of fresh sperm, equilibrium sperm at low temperature, and frozen sperm from goats. Utilizing the UHPLC-QTOF-MS technique combined with bioinformatics analysis, we screened a total of 65 DAMs and their correlations with 25 significant biological pathways implicated in sperm functions. Among them, one differential compound was enriched into one or more significant functional terms. The significant terms, such as biosynthesis of unsaturated fatty acids, glycine, serine and threonine metabolism, sphingolipid metabolism, and alpha-Linolenic acid metabolism, were captured. Along with other significant pathways, they may elucidate how sperm quality was affected by changing the temperature [28,29,30].

For instance, TCA cycle is an important metabolic route for satisfying the needs of bioenergy, biosynthesis and redox homeostasis in cellular mitochondria [31,32,33]. This cycle consists of multiple sugars, lipids, amino acid and intermediates. The intermediates include citric acid, isocitric acid, fumaric acid, L-malic acid, etc. Most of them belong to organic acids. Among them, citric acid is a tricarboxylic acid synthesized from acetyl coenzyme A (acetyl-CoA) and oxaloacetate, exerting a vital role in the metabolism of sugars, fats and amino acids [34]. Currently, citric acid is popular for use in animal feed as an additive because of its many positive effects; for example, it promotes phosphorus utilization, enhances beneficial gut microbiota, strengthens the intestinal tight junction barrier, and improves individual weight-gain by stimulating the growth of intestinal epithelial cells [35,36]. Isocitric acid, the isomer of citric acid, can be converted to citric acid by the action of aconitase, and also can be decarboxylated into α-ketoglutarate, which is an important signaling molecule in various metabolic processes [32]. Notably, it has been found that isocitric acid has two asymmetric carbon atoms that result in the appearance of four isomers, of which the isomer threo-Ds-isocitric acid has antistress, antihypoxia and antioxidant effects [37,38,39]. Fumaric acid is another carboxylic acid in the TCA cycle; it mainly serves as a source of intracellular energy in the form of ATP [40]. Also, this compound could scavenge intracellular ROS by affecting the glutathione redox cycle [41]. In view of its simple structure and roles, fumaric acid has been widely used as a positive additive in foods and animals feed, and it has no side effects [40,42]. For example, it has been reported that the replenishment of fumaric acid with 2.15 mM into extenders could significantly improve the acrosome integrity and mitochondrial membrane functionality of frozen goat sperm [43]. L-malic acid is biosynthesized from fumaric acid in the TCA cycle. The dehydrogenation process of L-malic acid is accompanied by the production of NADH, which is a crucial source of ATP [44]. Furthermore, L-malic acid has other capacities, such as antibacterial and antioxidant effects, affecting lipid metabolism [44,45,46]. In the current study, we found that the above four important intermediates were abundant in goat sperm. Compared to fresh sperm, citric acid and isocitric acid were upregulated in frozen sperm, while fumaric acid and L-malic acid were upregulated in equilibrium sperm, which implied the contribution of these compounds to sperm energy metabolism under specific conditions.

Unsaturated fatty acid (UFA) is a kind of indispensable fatty acid in tissues and cells [47,48]; it is classified into monounsaturated fatty acids (MUFAs), ω-3 polyunsaturated fatty acids (ω-3 PUFAs), and ω-6 polyunsaturated fatty acids (ω-6 PUFAs) based on the number, location, as well as functional characteristics of double carbon bonds [49]. Among these, PUFAs are important components of cell membrane phospholipids, which bestow the ideal biophysical properties upon the membrane [50]. In sperm cells, studies have shown the importance of PUFAs in maintaining membrane fluidity and permeability, resisting damage caused by ice crystal formation and lipid peroxidation, inhibiting inflammatory processes, and improving sperm viability [39,51,52], a lot of which may be attributed to its molecular properties with both hydrophilic and hydrophobic groups [53]. Predominant PUFAs such as linoleic acid (LA), arachidonic acid (AA) and docosahexaenoic acid (DHA) may affect male fertility via alterations in semen parameters and sperm function [51,54]. Specifically, LA is the parent compound of ω-6 PUFAs. Like other UFAs, its primary role is to maintain membrane fluidity as a structural component. In addition, when it is released from membrane phospholipids, it could also be enzymatically oxidized into various derivatives involved in cell signaling [55]. Our study shows that LA is abundant at higher levels in frozen goat sperm compared to fresh sperm, and it is significantly enriched in the biosynthesis of the unsaturated fatty acids pathway, which implies the importance of LA in sustaining the membrane structure and mobility of frozen sperm. On the other hand, AA can be released from membrane phospholipids via the activation of phospholipase A2 (PLA2) in abnormal sperm. The high content and abnormal metabolism of AA causes it to convert into a range of metabolites as an effect of different enzymes, which may induce lipid peroxidation. Its products have been considered as the direct executioners of ferroptosis—an iron-dependent nonapoptotic form triggered by an insufficiency of cellular antioxidases [56,57], which is detrimental to sperm motility and fertility [58,59]. Undeniably, equilibrium and cryopreservation processes cause negative effect on sperm motility. Taken together, this could explain why AA is upregulated in equilibrium sperm and frozen sperm compared to fresh sperm. DHA, an important member of ω-3 PUFAs, been reported to be downregulated in sheep sperm during cooling equilibrium and cryopreservation [60], but we found the opposite result in goat sperm. Similarly to our findings, 10–30 ng/mL DHA supplementation yielded the opposite result in extenders, causing damage to the mitochondrial activity, rather than improving the seminal quality indices in frozen goat sperm [61]. Historically, DHA was thought to be beneficial for testis development and spermatogenesis [62], and was thought to improve sperm motility via altering membrane lipid composition [63,64]. The effects of this compound on frozen sperm may be related to the species and the content [65], which requires further investigation.

Sphingolipids are structural components that make up ~10–20% of total membrane lipids, and are designed to maintain the membrane integrity in eukaryotic cells [66]. In addition, sphingolipids also act as bioactive molecules involved in some signaling pathways, such as cell proliferation, apoptosis, and inflammatory response, to facilitate cell survival, development and maturation [67,68,69]. Following three comparisons, our study identified nine differential sphingolipid compounds in goat sperm related to sphingolipid metabolism; the most important lipids, such as 3-dehydrosphinganine, L-serine and sphinganine, were upregulated in equilibrium sperm and frozen sperm. It has been reported that 3-dehydrosphinganine can be used as a biomarker for male sex determination [70], which could be synthesized from serine [71]. Moreover, L-serine serves as an important precursor for the synthesis of sphingolipids [72,73]. Sphinganine phosphorylation is necessary for producing a bioactive lipid mediator, sphingosine-1-phosphate, which plays a vital role in male germ cell anti-apoptosis [74,75]. In this light, the upregulation of the three substances was found to be beneficial to sperm during cryopreservation, and they may be used as potential biomarkers for evaluating sperm quality and fertility. 

Glycine, serine and threonine metabolism is one of most important pathways in sperm; it is associated with mitochondria protection, motility maintenance and acrosome reaction [28,76,77]. In goat sperm, we identified seven differential compounds, such as betaine, choline and L-cystathionine, that were significantly different among three comparisons, and which were enriched in glycine, serine and threonine metabolism. Betaine is a non-toxic, pleiotropic natural compound that is widely present in organisms [78,79]. In sperm, betaine has been found to enhance oxidation resistance by increasing mRNA levels in antioxidant-related genes such as Sod1, Gpx1 and Cat, and by inhibiting lipid peroxidation [80]. Additionally, betaine could maintain sperm motility and plasma membrane integrity during the freezing and thawing process [79]. Our study suggested that betaine was downregulated in frozen sperm and associated with glycine, serine and threonine metabolism. Choline is an antioxidant-based micronutrient involved in glycine, serine and threonine metabolism in living cells [81]; it can also act as a precursor of multiple metabolites like sphingomyelin, neurotransmitter acetylcholine, and the methyl donor of betaine. As such, the oxidation of choline to betaine is accompanied by the production of electron transport chain substrates, which result in the generation of ATP molecules in sperm mitochondria [82,83], so the downregulation of and deficiencies in choline are detrimental to sperm motility. L-cystathionine (L-Cth) is also an important metabolite of the cellular antioxidant family; for example, it can scavenge superoxide anion, NO and H_2_O_2_, enhance activities of GSH-Px, SOD and CAT, and concurrently protect DNA from damage [84,85]. We deduce from these facts that L-Cth may be involved in sperm antioxidation, thus serving sperm motility during the equilibrium. On the contrary, our study identified 2-ketobutyric acid in goat sperm for the first time, which is enriched in glycine, serine and threonine metabolism, and upregulated in equilibrium and frozen spermatozoa. This compound has been used in the food industry because of its safety and ability to destroy flavacin by damaging its cell membrane [86], but its specific effects on mammalian sperm cells need to be further explored. 

α-Linolenic acid (ALA) belongs to ω-3 PUFAs and is needed for organism health; it mostly produces energy through β-oxidation, but a small part can also be converted into DHA and eicosapentaenoic acid [87]. Studies have reported that ALA has multiple effects, such as antioxidizing, anti-inflammatory, and regulation of the intestinal flora. Regarding its antioxidizing effects, previous experiments suggest that 15–45 μg/mL ALA increases the activities of CAT and SOD, and simultaneously reduces contents of MDA and NO in human lymphocytes [88]. Moreover, 15 ng/mL ALA effectively inhibits ROS production in chlorambucil-induced early endothelial cells in rats [89], while 0.5–1.0% ALA in total lipids can reduce methylmercury-induced oxidative damage by enhancing the activities of SOD, CAT, GSH and GPx [90,91]. In sperm, Lee et al. found that an excess of free fatty acids, including linolenic acid, oleic acid and linoleic acid, is toxic to sperm, which is one of the physiological factors affecting sperm motility and spontaneous aggregation [92]. On the other hand, 5 ng/mL ALA replenishment can improve frozen bovine sperm motility, as well as the integrities of acrosome and plasma membrane [93], while 3 ng/mL ALA combined with carrier proteins is beneficial for increasing membrane integrity and mitochondrial activity by inhibiting lipid peroxidation [94]. Our results indicate that goat sperm exists naturally in ALA metabolism, while alterations of metabolites such as 13(S)-HpOTrE, trans-2-Enoyl-OPC8-CoA and (+)-7-isojasmonic acid CoA are associated with ALA metabolism during the sperm-freezing process. Among these, 13(S)-HpOTrE is one of the 15-lipoxygenase metabolites of ALA, and has been shown to play dual roles in anti-inflammation and apoptosis inhibition in cells [95], which suggests that a downregulation of 13(S)-HpOTrE in frozen sperm may be positively correlated with sperm viability.

Pyruvate metabolism has a typically bidirectional effect on sperm energy production, which can be influenced by lactate oxidation and is accompanied by NADH generation to provide electrons for the electron transport chain in the mitochondria; also, the reduction of pyruvate to lactate with the concomitant provision of NAD+ enhances the efficiency of glycolysis in the flagellum that produces ATP [96,97]. Noticeably, the cryopreservation process changes some compounds’ abundance, making them enriched in sperm pyruvate metabolism. For example, L-lactic acid, D-lactic acid, fumaric acid, and L-malic acid metabolites were upregulated in equilibrium sperm and frozen sperm compared to fresh sperm. L-lactic acid and D-lactic acid are enantiomers of lactic acid, while L-lactic acid is a common compound in metabolic pathways, and D-lactic acid is a harmful enantiomer in mammals [98]. Given our results, it can be speculated that L-lactic acid may be involved in the oxidation reaction and generate NADH for electrons supply to the electron transport chain in the mitochondria; concurrently, D-lactic acid may have an inhibitory effect on sperm motility during the freezing process [99], and the specific mechanisms merit further exploration. Additionally, fumaric acid and L-malic acid have been discussed previously as potential sources of ATP formation [40,44]. In sum, the results imply the effects of these compounds on sperm energy metabolism under specific conditions.

## 4. Materials and Methods

### 4.1. Study Design and Sample Collection

In the current study, an ultra-high-performance liquid chromatography coupled with quadrupole time-of-flight mass spectrometer (UHPLC-QTOF-MS) technology was applied to identify and characterize the metabolite profiles in goat sperm (Appendix A). The sperm samples were obtained from Kunming Yi Xingheng Animal Science and Technology Co., Ltd. (located at 102°37′ East and 24°57′ North, Kunming, China) and the experiments did not involve the slaughter of live animals.

Ten healthy adult *Yunshang black* goats (approximately 2–3 years of age, with average live body weights of 84.00 ± 2.38 kg) were selected randomly. This breed is the first meat black goat breed in China, which has been bred successfully by our institute in cooperation with adjacent institutes for 22 years. It is recognized for its multiple excellent characteristics, such as its fast growth rate and early maturity, high levels of meat production and satisfactory meat quality, year-round estrus, prolificacy, and strong environmental adaptability. At present, this breed plays an important role in the development of animal husbandry in the south of China. Before experiments, the above ten males were housed under the same management system. Fresh ejaculates (two ejaculates per male in 10 min) were collected using an artificial vagina. Then, the collected ejaculates of each male were pooled and mixed with the protease inhibitors (including 104 mM 4-(2-Aminoethyl) benzenesulfonyl fluoride hydrochloride, 80 μM aprotinin, 5 mM bestatin, 1.5 mM proteinase inhibitor E-64, 2 mM leupeptin and 1.5 mM pepstatin A in dimethyl sulfoxide, sparkJade, Jinan, China), and placed in a water bath at 37.0 °C. The semen quality was immediately assessed using the computer-assisted sperm analysis system (CASA, Microptic, Barcelona, Spain); only ejaculates with a minimum volume of 1.0 mL, motility greater than 75% and a concentration greater than 3 × 10^9^/mL could be used for further analysis [100,101]. The pooled semen of each male was subsequently divided into three aliquots. One was used as a fresh sperm source (control sperm sample, C group), one was used as a cooling sperm source (equilibrated sperm sample at low temperature, E group), while the remaining one was used for cryopreservation (frozen–thawed sperm sample, F group).

### 4.2. Cooling and Freeze-Thaw Processing

The above two semen aliquots (E and F groups) were gently diluted using the same medium at RT (25 °C) at a final sperm concentration of about 3 × 10^8^/mL, respectively [102]. The medium was made up of 20% (*v*/*v*) AndroMed^®^ (Minitube, Tiefenbach, Germany), 80% (*v*/*v*) double-distilled water, 5 mg/mL lactose (Sigma-Aldrich, Saint Louis, MO, USA), and 3 µg/mL vitamin B12 (Cisen Pharmaceutical, Jining, China) [6]. The two extended semen samples were then loaded into the labeled 0.25 mL straws (IMV technologies, L’Aigle, France) and equilibrated at 4 °C for 3.5 h. For the F group sample, after equilibriation at 4 °C, it was placed in liquid nitrogen vapor for 8 min (−80 °C); the distance was 5 cm between straws and the liquid nitrogen level. Then, all straws were immediately frozen in liquid nitrogen for a week [8,103]. For the thawing process, the frozen specimen was incubated in a water bath at 37 °C for 35 s.

After cooling and the freeze–thaw processing, all semen samples were centrifuged at 10,000× *g* for 3 min at 4 °C, and the pellets were washed thrice before being subjected to omics analysis [8,104].

### 4.3. Extraction of Metabolite

To exact sperm metabolites, 50 mg of each sperm pellet was initially dissolved in 200 µL of precooled water, followed by the addition of 800 μL of a precooled methanol/acetonitrile solution (1/1, *v*/*v*). This mixture was then subjected to vortexing and ultrasonic disruption in an ice bath for 1 h using an ultrasonic cell crusher (Biosafer, Orlando, FL, USA). A 50 µL aliquot of the homogenate solution was utilized to determine protein concentration using a BCA kit (Beyotime, Shanghai, China), following the manufacturer’s instructions. The remaining mixture was further processed by adding 500 µL of acetonitrile/methanol (1:1) solvent, vortexed for 30 s, and incubated at −20 °C for 1 h to facilitate protein precipitation. This was followed by centrifugation at 16,000× *g* for 20 min at 4 °C [105]. The resulting 500 μL supernatant was then transferred to a new tube, dried in a vacuum concentrator, and reconstituted in 1 mL of extraction solvent (methanol/acetonitrile/water, 2/2/1). After additional vortexing and ultrasonic disruption for 10 min, the sample was centrifuged at 16,000× *g* for 15 min at 4 °C. The clear supernatant was subsequently filtered through 0.22 μm microfilters and transferred to a sample vial for LC-MS analysis [6]. Quality control (QC) samples were prepared by mixing equal aliquots of the supernatant from all sperm samples, also named mix samples. The QC samples were used to normalize the data for metabolic profiling. During instrumental analysis, one QC sample was usually inserted into every 15 testing samples to monitor the repeatability of the analysis process.

### 4.4. UHPLC-QTOF-MS

Metabolomics analyses were performed using an UHPLC-QTOF-MS system (Agilent Technologies, Santa Clara, CA, USA). A 2.1 × 100 mm ACQUITY UPLC HSS T3 1.8 μm column was used to analyze samples with a flow rate of 0.3 mL/min at 35 °C, and an injection volume of 1.0 μL. Mobile phases A (0.04% formic acid in water) and B (acetonitrile) were selected for UPLC chromatography. The gradient profile of phase B started at 5% for 0–2 min, increased to 70% for 9 min, then climbed to 90% for 10 min and was held at 90% for 2 min, before being reduced back to 5% for 0.01 min and held at 5% for an additional 2 min. For the acquisition of MS and MS2 data, a 6545Q-TOF mass spectrometer was conducted for electrospray ionization (ESI), with the specific parameters set as follows: sheath gas flow rate 11 Arb, auxiliary gas flow rate 8 Arb, spray temperature 325 °C, full scan mass range *m*/*z* 50–1200 Da, and nebulizer pressure 40 PSI. The spray voltage was set at 0.25 kv for the positive mode and 1.50 kv for the negative mode, respectively [106,107]. For LC-MS/MS data processing, the raw data files were first converted into mzML format by the ProteoWizard software (http://proteowizard.sourceforge.net (accessed on 18 August 2024), version 3.0.9134). Peak extraction, alignment and retention time (RT) correction were performed by the XCMS program, and the peak area was corrected by the “SVR” method. For extracted data, ion peaks with missing values ≤ 50% in each group of samples were retained for subsequent statistical analysis. The total peak areas of positive and negative ion data were normalized, respectively; the peaks of positive and negative ions were integrated and pattern recognition was carried out with the R program. The metabolites were identified by accuracy MS matching (mass tolerance < 20 ppm) and MS2 matching (mass tolerance < 0.02 Da), and were matched with the public HMDB database (https://hmdb.ca/, access date was 18 August 2024) and metDNA software (http://metdna.zhulab.cn/, access date was 18 August 2024), combined with a self-built metabolite standard library.

### 4.5. Statistical Analysis

Omics data were statistically analyzed by the R project (https://cran.r-project.org/, 18 August 2024, version 3.6.1), as well as Student’s *t*-test combined with principal component (PCA) and orthogonal Partial least square discriminant (OPLS-DA) analyses. Functional enrichment analysis was also conducted using the KEGG database (https://www.kegg.jp/, access date was18 August 2024), with Fisher’s exact testing and False Discovery Rate correction for multiple testing, respectively. Enriched KEGG pathways were deemed statistically significant at a corrected *p*-value of less than 0.05 [108]. The statistical approaches ensured a rigorous assessment of the data, allowing for meaningful and reliable interpretations of the study’s findings.

## 5. Conclusions

In summary, our study generated metabolite profiles of goat sperm and their changes in abundance induced by the freezing and thawing process. These changes may be tightly associated with sperm cryoinjuries. The comprehensive bioinformatic analysis highlighted 65 DAMs as well as 25 significantly enriched pathways, which may be involved in the regulation of sperm characteristics and function. Our findings provide important insights into the molecular mechanisms of metabolism that influence the post-thaw quality and freezability of goat sperm. Additionally, as compared to the freezing and thawing process, the impact of equilibration at low temperature on sperm quality may require more attention. Furthermore, based on some of the key metabolite molecules we obtained, such as ceramide, betaine, choline, fumaric acid, L-malic acid, and L-lactic acid, exploring the positive effects of their supplementation in freezing extenders on the quality of frozen sperm is also worthy of study in future research.

## Figures and Tables

**Figure 1 ijms-25-09112-f001:**
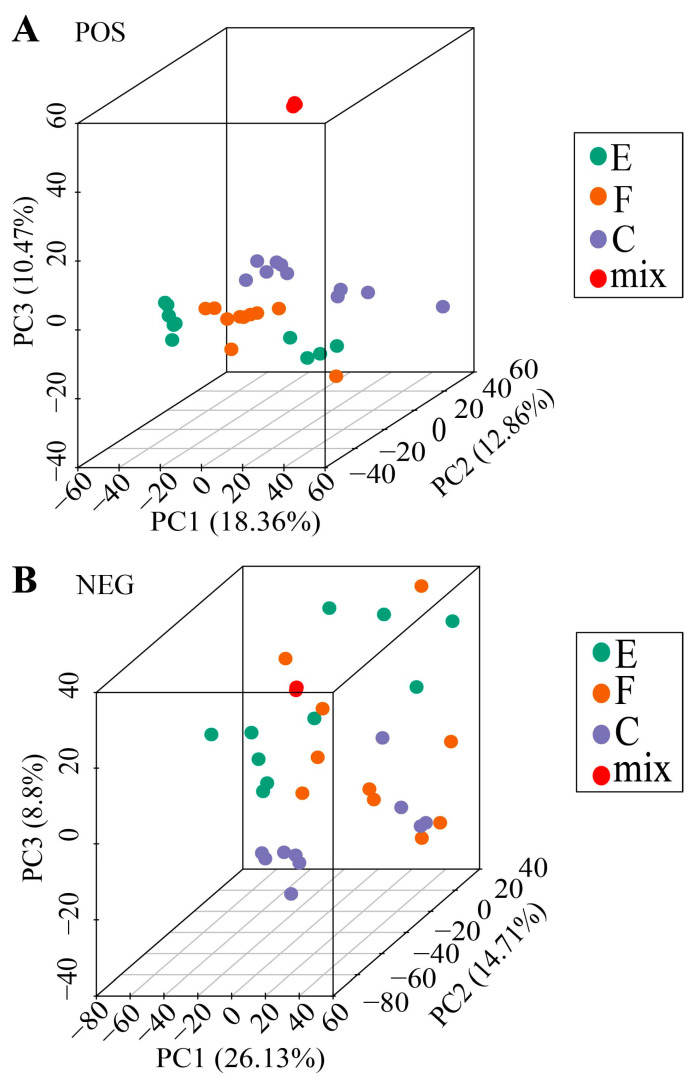
PCA 3D representation for QC assessment in POS (**A**) and NEG (**B**) modes, respectively.

**Figure 2 ijms-25-09112-f002:**
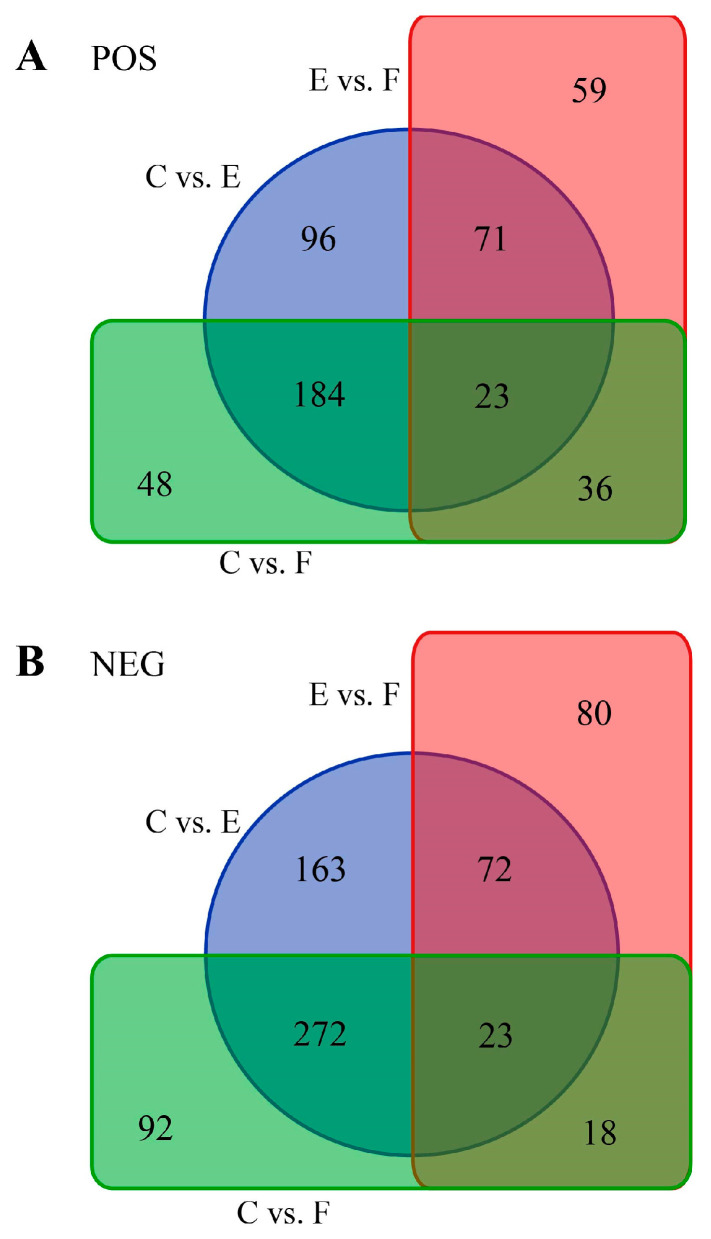
Venn diagram displaying numbers of DAMs among the three comparative groups in POS (**A**) and NEG (**B**) modes, respectively.

**Figure 3 ijms-25-09112-f003:**
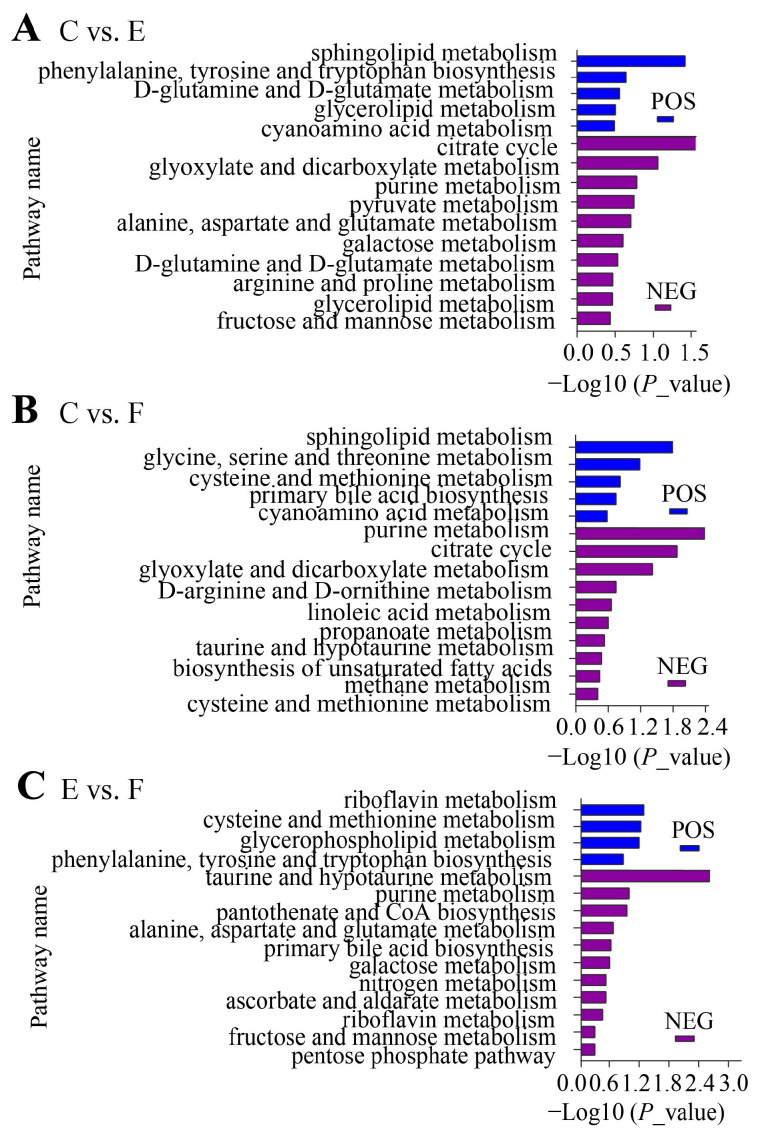
The DAMs were mainly involved in various biological pathways in C vs. E comparison (**A**), C vs. F comparison (**B**), and E vs. F comparison (**C**), respectively.

**Figure 4 ijms-25-09112-f004:**
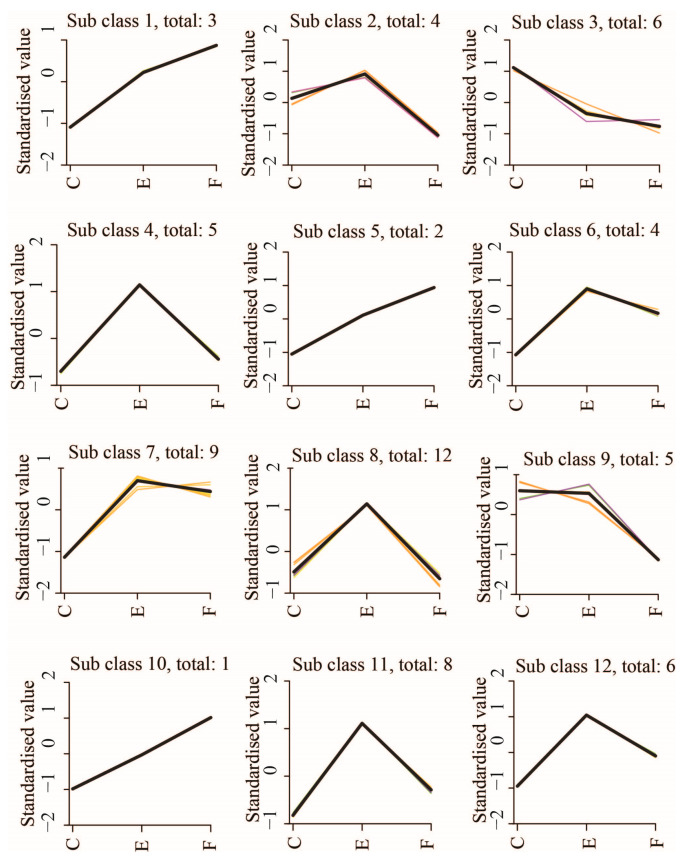
Distribution profiles of 65 significant DAMs by Mfuzz clustering. The trend of red and yellow lines indicates that there is a minor deviation between the abundance trends of a few DAMs and the abundance trends of most DAMs in this class.

**Figure 5 ijms-25-09112-f005:**
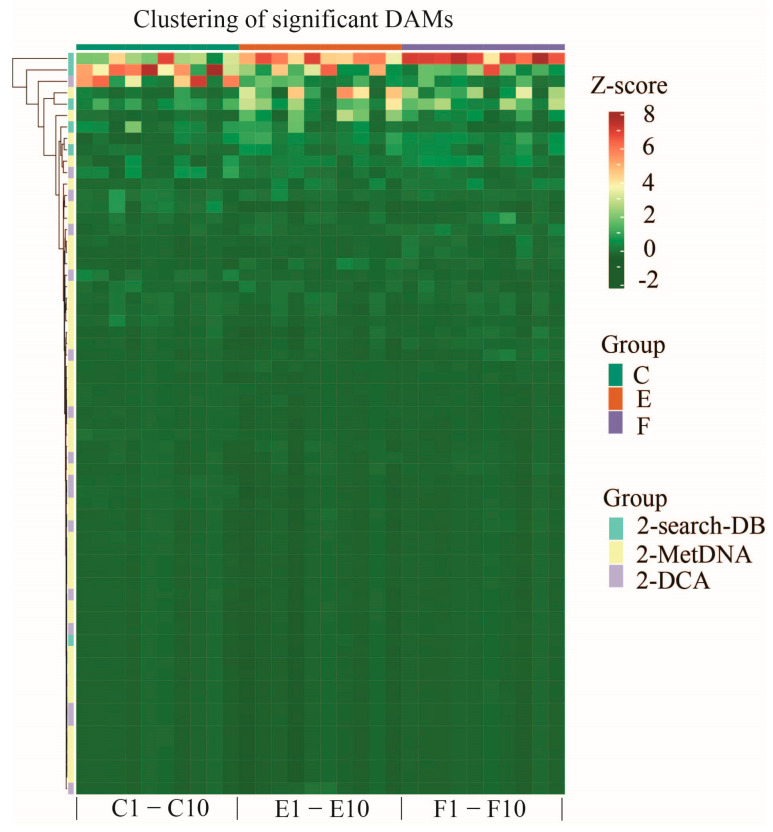
Visualized heat map of hierarchical clustering for 65 significant DAMs among the three comparative groups in positive and negative modes, and these DAMs were mainly categorized into 2-search-DB, 2-MetDNA, and 2-DCA classes.

## Data Availability

Data are contained within the article and Appendix A.

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
