# Peer review of "Revisiting the Injury Mechanism of Goat Sperm Caused by the Cryopreservation Process from a Perspective of Sperm Metabolite Profiles"

_ijms, 2024, doi:10.3390/ijms25169112_

Round 1

Reviewer 1 Report

Comments and Suggestions for Authors

I would like to congratulate the authors for the work they have done. It is very important to focus on the molecular mechanisms that affect gamete manipulation processes, which may end up affecting not only pregnancy rates but also the viability of the newborn. I consider the article to be well-posed, but it is not very enriching since it is limited only to the data obtained from a metabolomics. I consider that this information is publishable but it is not very useful since it does not offer global information about what is happening. It would be better to compare these results with a proteomics, to do a study of target proteins that you think may be affected, for example, by WB and even to try to reverse with some strategy the effects observed in metabolomics after refrigeration and freezing.

Regarding results:

Figure 1: what is mix? I cannot find it in the text and the legend does not explain anything.

Figure 2 is not clarifying, the diagram is very confusing since it is not clear which metabolites belong to each group. The legend of the figure also fails to describe the figure correctly. A legend should allow the figure to be understood without having to resort to text. It needs to be improved.

Figure 5: it also does not have a well-explained legend. The results section also lacks an explanation of what happens in this heat map.

The discussion is extensive but it is fine.

In terms of methodology, a more detailed explanation of how KEGGs are obtained is needed.

Author Response

Respond to Reviewer1's comments:

General comments: [I would like to congratulate the authors for the work they have done. It is very important to focus on the molecular mechanisms that affect gamete manipulation processes, which may end up affecting not only pregnancy rates but also the viability of the newborn. I consider the article to be well-posed, but it is not very enriching since it is limited only to the data obtained from a metabolomics. I consider that this information is publishable but it is not very useful since it does not offer global information about what is happening. It would be better to compare these results with a proteomics, to do a study of target proteins that you think may be affected, for example, by WB and even to try to reverse with some strategy the effects observed in metabolomics after refrigeration and freezing.]

General response:

Dear reviewer 1, firstly, thank you so much for your in-depth comments concerning our manuscript, which are all valuable and very helpful for revising and improving our paper!

With regard to the meaningful points in above general comments, we fully agree. In the later studies, we will further explore the molecular spectrum changes like proteome and lipidome in sperm during the freezing-thawing process, and identify the molecular abundance changes closely related to sperm characteristics and function, as well as their enriched important molecular pathways, simultaneously, perform multi-omics association analysis to offer global information about the influence mechanisms of the freezing-thawing process on goat sperm quality and its' fertility.

In addition, following those comments and suggestions about this paper, we have made correction carefully in the manuscript. Now, those revised portion were marked up using the “track changes with red fonts” in this pdf file.

About minor comments:

Comments_1: [Figure 1: what is mix? I cannot find it in the text and the legend does not explain anything.]

Response_1: Thanks for your question. Mix represents quality control (QC) sample. The QC sample is prepared by mixing extracts from all testing samples to analyze the repeatability of samples under the same treatment method. During instrumental analysis, one QC sample is usually inserted into every 15 testing samples to monitor the repeatability of the analysis process. About the mix, we've replenished corresponding description for it in the text, pleased see lines 360-364 in the “Revised manuscript”.

Comments_2: [Figure 2 is not clarifying, the diagram is very confusing since it is not clear which metabolites belong to each group. The legend of the figure also fails to describe the figure correctly. A legend should allow the figure to be understood without having to resort to text. It needs to be improved.]

Response_2: Thank you for your reminding. We have refined the legend for Figure 2. Figure 2 was a venn diagram that displaying the numbers of DAMs in POS (A) and NEG (B) modes, respectively. We think this graph was clear in terms of showing the numbers' distribution of DAMs from each comparative group in both modes. Specifically, there were 374 DAMs (96+71+184+23=374 in graph) in C vs. E, 291 DAMs (48+184+23+36=291 in graph) in C vs. F, and 189 DAMs (59+71+23+36=189 in graph) in E vs. F in the positive mode (POS), which showed in Figure 2A; concurrently, there were 530 DAMs (163+72+272+23=530 in graph) in C vs. E, 405 DAMs (92+272+23+18=405 in graph) in C vs. F, and 193 DAMs (80+72+23+18=193 in graph) in E vs. F in the negative mode (NEG), which showed in Figure 2B. In addition, we also performed corresponding description for these results in the text, pleased see lines 89-94 in the “Revised manuscript”.

Comments_3: [Figure 5: it also does not have a well-explained legend. The results section also lacks an explanation of what happens in this heat map.]

Response_3: Thank you for your reminding. We've refined the legend for Figure 5, at the same time, we've replenished corresponding description for these results in the text, pleased see lines 109-119 in the “Revised manuscript”.

Comments_4: [In terms of methodology, a more detailed explanation of how KEGGs are obtained is needed.]

Response_4: Thanks. We've checked the part of ‘Materials and Methods’ in the text carefully, and have replenished description about ion peaks filtration, data matching and metabolite identification, please see lines 380-388. Concurrently, we've also added details about KEGG analysis, please see lines 392-395 in the “Revised manuscript”.

Thanks a lot again for your help and comments!

Reviewer 2 Report

Comments and Suggestions for Authors

In the present study, the authors answer several important scientific questions related to the topic of goat sperm cold,- and cryo-injury mechanisms using sophisticated omics analysis. The manuscript fits the general scope of the journal and SI. Section "Introduction" provides background information sufficiently. Section "Material and Methods" contains all the necessary details so the experiments conducted in the present study are repeatable. "Results" and "Discussion" present and interpret the obtained data well. The conclusion of the study is supported by the results. Overall, the novelty of the study is high, with no significant flaws; the manuscript needs only a minor revisions. Therefore I recommend accepting the manuscript for publication.

However, I have several comments to the authors:
1) Fig.1: I am not sure that the presentation of 3D figures on the 2D plane is good for presentation. I recommend (if possible) to redraw the figure to fit the 2D plane.
2) I recommend to include the list of abbreviatures.
3) Please, give some more details about the goat breed used in the study.
4) Line 305: please, give details about the protease inhibitors (concentration, etc).
5) Line 321: Please, describe the freezing process in more detail (namely, what was the distance between straws and LN?)
6) I recommend including in the conclusion one or two paragraphs with any practical recommendations on the addition of some compounds to the extender with the purpose of mitigating of the negative effect of cold,- and cryo-injury, based on the results, obtained in the present study.

I thank the authors for their valuable contribution. 

Author Response

Respond to Reviewer2's comments:

General comments: [In the present study, the authors answer several important scientific questions related to the topic of goat sperm cold, - and cryo-injury mechanisms using sophisticated omics analysis. The manuscript fits the general scope of the journal and SI. Section "Introduction" provides background information sufficiently. Section "Material and Methods" contains all the necessary details so the experiments conducted in the present study are repeatable. "Results" and "Discussion" present and interpret the obtained data well. The conclusion of the study is supported by the results. Overall, the novelty of the study is high, with no significant flaws; the manuscript needs only a minor revision. Therefore, I recommend accepting the manuscript for publication.]

General response:

Dear reviewer 2, firstly, thank you very much for your good comments concerning our manuscript, which are all valuable and very helpful for revising and improving our paper! Following those comments and suggestions, we have made correction carefully in the manuscript. Now, those revised portion were marked up using the “track changes with red fonts” in this pdf file.

About minor comments:

Comments_1: [Fig.1: I am not sure that the presentation of 3D figures on the 2D plane is good for presentation. I recommend (if possible) to redraw the figure to fit the 2D plane.]

Response_1: Thanks for your suggestion. We think that 3D PCA could provide richer dimension and more detailed information as compared to 2D PCA, so we showed the 3D PCA plot in Fig. 1. In addition, we've also replenished “2D PCA plot (.pdf version)”, which showed basic structure and trends of the data, please refer it.

Comments_2: [I recommend to include the list of abbreviatures.]

Response_2: Thanks for your suggestion. We've provided Abbreviation List abbreviatures list, please see line 411 in the “Revised manuscript”.

Comments_3: [Please, give some more details about the goat breed used in the study.]

Response_3: Thank you for your suggestion. We've replenished more details about the goat breed used in this study, please see lines 310-317 in the “Revised manuscript”.

Comments_4: [Line 305: please, give details about the protease inhibitors (concentration, etc).]

Response_4: Thanks. We've added composition and corresponding concentration of protease inhibitors, please see lines 320-322 in the “Revised manuscript”.

Comments_5: [Line 321: Please, describe the freezing process in more detail (namely, what was the distance between straws and LN?)]

Response_5: Thank you for your reminder. When freezing, sperm contained straws were placed in liquid nitrogen vapor for 8 min (-80°C), namely, the distance was 5 cm between straws and liquid nitrogen level. Then, all straws were immediately frozen in liquid nitrogen for a week. We've replenished these informations into the text, please see lines 338-341 in the “Revised manuscript”.

Comments_6: [I recommend including in the conclusion one or two paragraphs with any practical recommendations on the addition of some compounds to the extender with the purpose of mitigating of the negative effect of cold,- and cryo-injury, based on the results, obtained in the present study.]

Response_6: Thanks. We've added corresponding content in the ‘Conclusions’, please see lines 407-410 in the “Revised manuscript”.

Thanks again! We have revised this manuscript according to your suggestions. Please see the sentences marked with red fonts in the “Revised manuscript”.
